# Photoactivatable ribonucleosides mark base-specific RNA-binding sites

Jong Woo Bae [1,2], Sangtae Kim[3], V. Narry Kim [1,2✉] & Jong-Seo Kim [1,2✉]

RNA-protein interaction can be captured by crosslinking and enrichment followed by tandem mass spectrometry, but it remains challenging to pinpoint RNA-binding sites (RBSs) or provide direct evidence for RNA-binding. To overcome these limitations, we here developed pRBS-ID, by incorporating the benefits of UVA-based photoactivatable ribonucleoside (PAR; 4-thiouridine and 6-thioguanosine) crosslinking and chemical RNA cleavage. pRBS-ID robustly detects peptides crosslinked to PAR adducts, offering direct RNA-binding evidence and identifying RBSs at single amino acid-resolution with base-specificity (U or G). Using pRBS-ID, we could profile uridine-contacting RBSs globally and discover guanosine-contacting RBSs, which allowed us to characterize the base-specific interactions. We also applied the search pipeline to analyze the datasets from UVC-based RBS-ID experiments, altogether offering a comprehensive list of human RBSs with high coverage (3,077 RBSs in 532 proteins in total). pRBS-ID is a widely applicable platform to investigate the molecular basis of posttranscriptional regulation.

[1] Center for RNA Research, Institute for Basic Science, Seoul 08826, Korea. [2] School of Biological Sciences, Seoul National University, Seoul 08826, Korea. [3] Seer Inc., Redwood City, CA 94065, USA. ✉email: narrykim@snu.ac.kr; jongseokim@snu.ac.kr

RNA-binding proteins (RBPs) are integral players in post-transcriptional gene regulation[1–3]. To understand their roles and action mechanisms in gene regulation, early pioneering studies focused on cataloging RBPs using RNA interactome capture methods[4,5]. RBPs were crosslinked to RNA using UV, pulled down in an RNA-dependent manner, and analyzed by liquid chromatography and tandem mass spectrometry (LC-MS/MS). Adding onto the initial protein-level mapping of RBPs, recent studies narrowed down to systematically profile RNA-binding domains (RBDs) within RBPs[6–8]. In these studies, the RBDs were inferred from the depletion of RNA-crosslinked peptides in the whole proteome[7], or the enrichment of partially digested peptides adjacent to the actual RNA-crosslinked ones[6,8]. However, both methods heavily relied on quantitative enrichment analysis for RBS identification, which could only provide indirect RNA-binding evidence. Also, the resolution was low because the analyses were confined to peptide-level.

To overcome these limitations, additional LC-MS/MS-based methods were developed to identify the peptides that are directly crosslinked to RNA[9–14]. Briefly, the crosslinked RNA adducts were shortened as much as possible using extensive nuclease digestion. Then, peptides with the crosslinked RNA fragments were detected by MS, and the mass increased owing to the adduct was used as evidence for RNA binding. In this approach, the homogeneity of the shortened RNA moiety was crucial, yet nucleolytic digestion leaves behind a rather heterogeneous pool of RNA adducts. This complexity increases MS/MS search space, making it difficult to determine RNA-binding sites (RBSs) within peptides and severely reducing the coverage.

Addressing this concern, we recently implemented hydrofluoride (HF) to completely digest the crosslinked RNA adduct to a single ribonucleoside (RBS-ID[15]). The single ribonucleoside adduct minimized MS/MS search space, thereby drastically improving the RBS identification rate. This approach using chemical cleavage instead of enzymatic digestion enabled us to reliably pinpoint individual amino acids as RBSs, providing a single amino acid-resolution.

The original protocol of RBS-ID was developed to map UVC-crosslinked RBSs. UVC (254 nm) has been widely utilized to capture RNA−protein interactions, yet it holds substantial methodological drawbacks. UVC irradiation induces a covalent bond mostly between Us and amino acids in close proximity while the other RNA bases (A, G, or C) are crosslinked at low frequencies[16]. The crosslinked non-U bases could be PCR-amplified and detected by sequencing[16], but the peptides cannot be amplified so they remain below the detection limit of LC-MS/MS[9–15]. Thus, RBS identification using UVC-crosslinking has been limited to U-interactions, leaving out the other bases in RNA. Moreover, the low efficiency of UVC-crosslinking[17] made it difficult to identify low-abundance RNA−protein interactions from small amounts of samples. Thus, although RBS-ID could drastically expand the RBS maps, its application is still limited to abundant proteins, U-base, and amino acid types sensitive to UVC irradiation.

Photoactivatable ribonucleoside (PAR)-specific UVA-crosslinking is a promising alternative that can relieve the aforementioned shortcomings[17–21]. UVA irradiation (365 nm) selectively activates RNA-incorporated PARs to crosslink RBPs. The most widely utilized PARs are 4-thiouridine (4SU) and 6-thioguanosine (6SG)[17–21], which can expand the base-interaction repertoire of RBSs from single to double base types (U and G) with higher crosslinking efficiencies compared to their respective natural RNA base counterparts[17]. Selective PAR labeling and efficient UVA-crosslinking would be particularly useful for the mass spectrometric detection and identification of peptides crosslinked to individual RNA base types even at low abundances. Two previous studies attempted to map RBSs crosslinked to 4SU using nucleases to cleave the RNA moiety crosslinked to peptides in yeast cells[9,13], bringing many interesting insights to the molecular interactions between RNA and proteins. Yet, the methods utilized nucleases to digest peptide-crosslinked RNA, where the incomplete RNA cleavage may have reduced the resolution and coverage for RBS identification.

Here, we integrated the advantages of chemical RNA cleavage with PAR crosslinking. Initially, we uncovered the chemical features of HF-treated PAR adducts in MS/MS and built an MS/MS search pipeline specific to PAR-modification. Based on this, we developed pRBS-ID, a method that enables system-wide profiling of PAR-RBSs with a robust identification rate and accurate label-free quantification. Notably, we unraveled 4SU-RBSs in human cells with high coverage, along with enabling the identification of 6SG-RBSs. We further analyzed the domain and site-level characteristics of the PAR-RBSs and demonstrated the usefulness of pRBS-ID in the structural analysis of RNA−protein interactions.

## Results

**Characterization of PAR adduct moieties upon chemical RNA cleavage.** To identify the PAR-crosslinked RNA-binding sites, we needed to characterize the masses of PAR adducts on peptides. For this, HeLa cells were metabolically labeled with 4SU or 6SG and were irradiated with UVA (365 nm) to induce crosslinking between proteins and PARs in RNA. Following cell lysis, the protein-mRNA conjugates were enriched using oligo-d(T) beads[4,5], and digested with trypsin. Then, the peptide-RNA conjugates were enriched using a 30 kDa size-cutoff cellulose membrane filter to deplete non-crosslinked peptides smaller in size than the conjugates. The peptide-RNA conjugates were treated with hydrofluoride (HF) for complete RNA digestion[15], which leaves a single nucleoside or base conjugated to peptides, and subjected to LC-MS/MS analysis (Fig. 1a). The MS/MS data consists of two layers. MS1 measures the mass of a whole peptide precursor ion which is subsequently isolated in the gas phase and broken into smaller fragment ions, whose masses are measured in the MS2 stage.

First, we determined the PAR adduct masses on the peptide precursor ions. We utilized MSFragger[22] as an open search tool, which detects the precursor mass shifts in MS1 with little dependency on MS2 fragment ions (Supplementary Data 1). As a result, the base and nucleoside forms with $H_2S$ loss were identified as 4SU and 6SG adducts (Of note, hereafter 4-thiouracil/6-thioguanine are referred to as "4SU/6SG-base"; 4-thiouridine/6-thioguanosine as "4SU/6SG-nucleoside") (Fig. 1b, c, top). To understand how the PAR-base and PAR-nucleoside forms were generated, we carefully inspected the MS1 extracted ion chromatograms (XICs) of the same peptide with the respective adducts (Supplementary Fig. 1a, b). Among the XIC peaks corresponding to peptides with PAR-base adducts, some showed elution times distinct from the peaks from peptides with PAR-nucleoside. This indicates that HF digestion generates both the PAR-base and PAR-nucleoside forms in solution. Meanwhile, other XIC peaks from peptides with the PAR-base forms showed co-elution patterns with those of the nucleoside forms. This suggests that a portion of the PAR-nucleoside forms are in-source fragmented to the PAR-base forms during the ionization step prior to MS1 measurement. In addition, the loss of $H_2S$ is consistent with previous studies that used nuclease-based RNA digestion approaches[9,13], suggesting that $H_2S$ is lost independent of the HF treatment. It is likely that $H_2S$ loss occurs during UVA-induced protein-RNA crosslinking (Supplementary Fig. 1e).

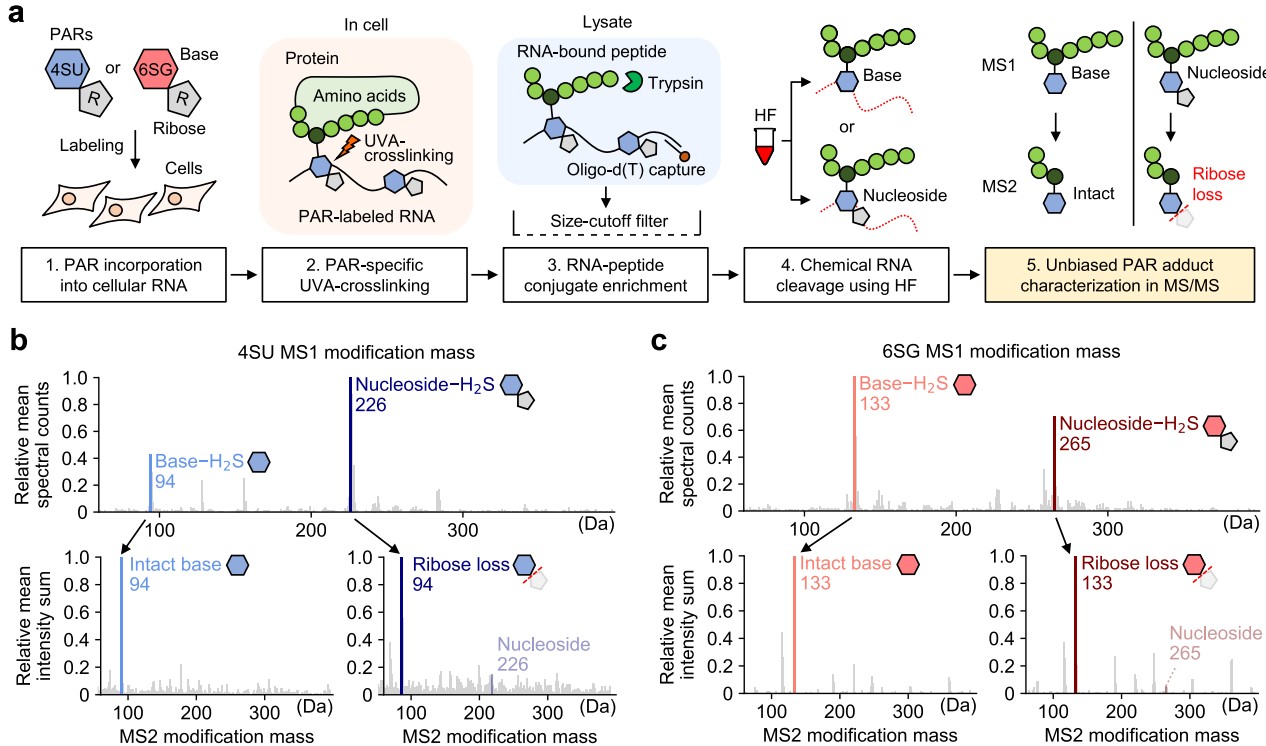

**Fig. 1 Characterization of HF-digested PAR adducts. a** Experimental scheme for sample preparation for pRBS-ID. Upon photoactivatable ribonucleoside (PAR) labeling, cells are irradiated with UVA to induce protein-RNA crosslinking. The protein-RNA conjugates are enriched using oligo-d(T)[4, 5], and the protein is digested into tryptic peptides. Then, the peptide-RNA conjugates are enriched by size-cutoff filtration, and HF is added to fully digest the RNA into monomers[15]. The resulting PAR adducts are characterized using two open search tools[22, 23]. This information serves as a basis for designing the MS/MS search pipeline for pRBS-ID. Open search for 4SU-RBS (**b**) or 6SG-RBS (**c**). MS1 modification mass was measured using MSFragger[22] (top), and the respective MS2 fragment ion modification on precursor ions with base (bottom left) or nucleoside (bottom right) adducts was analyzed, respectively. Modification mass ranges of 60−400 Da are shown.

Next, we further characterized the fragmentation behavior of PAR adduct masses in MS2 (Fig. 1b, c, bottom). Notably, we observed a nearly complete transition of the nucleoside form in the MS1-level to the base form in the MS2-level owing to the neutral loss of ribose, while the base form in the MS1-level remained intact in the MS2-level. This unique phenomenon was initially noticed in an open search using the MODa algorithm[23], which employs the adduct mass on MS2 fragment ions to find modifications in peptides (Supplementary Data 1). Here, we encountered an unexpected observation where the PAR-nucleoside forms were severely underestimated (Supplementary Fig. 1c, d), compared to the results obtained from MSFragger (Fig. 1b, c, top). This finding suggests partial or complete loss of the PAR-nucleoside modification during higher-energy collisional dissociation (HCD) for MS2 fragment ion generation. To accurately characterize the fragmentation patterns of PAR adducts in the MS/MS process, we carried out an in-depth analysis of the MS2 spectra from peptide-spectrum matches (PSMs) with MS1 adduct masses of PAR-nucleosides in MSFragger searches, matching the MS2 peaks to a list of all possible masses for MS2 fragment ions with adduct masses ranging from 1 to 400 m/z. For each nominal mass adduct, we collected well-matched MS2 fragment ions and summed up the MS2 peak intensities. As a result, we found that the nominal mass corresponding to the PAR-base form showed the largest intensity sum, indicating that the PAR-nucleoside form was further fragmented to the PAR-base form via partial neutral loss of ribose (Fig. 1b, c, each bottom right). When we further performed the same analysis on PSMs with MS1 adduct masses of PAR-bases, the most abundant nominal mass was found to be the same

(Fig. 1b, c, each bottom left). We additionally verified these observations by carefully inspecting individual MS2 spectra (Supplementary Figs. 2–4).

**Building an MS/MS search pipeline specific to PAR-modification.** Having determined the chemical features of PAR adducts in both the solution and gas phases (Supplementary Fig. 1e), we next designed an MS/MS search pipeline that can robustly profile the RBSs crosslinked to PARs. We focused on the observation that the modification masses at the MS2 stage from both solid (PAR-base form) and partially labile (PAR-nucleoside form, ribose neutral loss) modification were the same (PAR-base form). In the case of partially labile PAR-nucleoside modification, if the MS1 precursor mass is adjusted into a PAR-base adduct, the resulting combination of MS1 and MS2 ion masses would be identical to that of the solid PAR-base form modification. This MS1 correction would minimize the number of variable modifications to a single PAR-base and allow simultaneous identification of RBSs crosslinked to either PAR-bases or PAR-nucleosides.

Therefore, we integrated this idea into the MS/MS search pipeline design for pRBS-ID (Fig. 2a). Given that the MS/MS scans from PAR-nucleoside forms could not be predetermined, we decided to open up the possibilities for MS1 correction for all MS/MS scans. Following LC-MS/MS spectral data acquisition (Fig. 2a, step 1), each MS/MS scan data was duplicated. One was left unaltered, while the other's precursor mass was adjusted by subtracting the ribose mass, accounting for the ribose loss from the PAR-nucleosides (Fig. 2a, step 2). Then, the duplicated

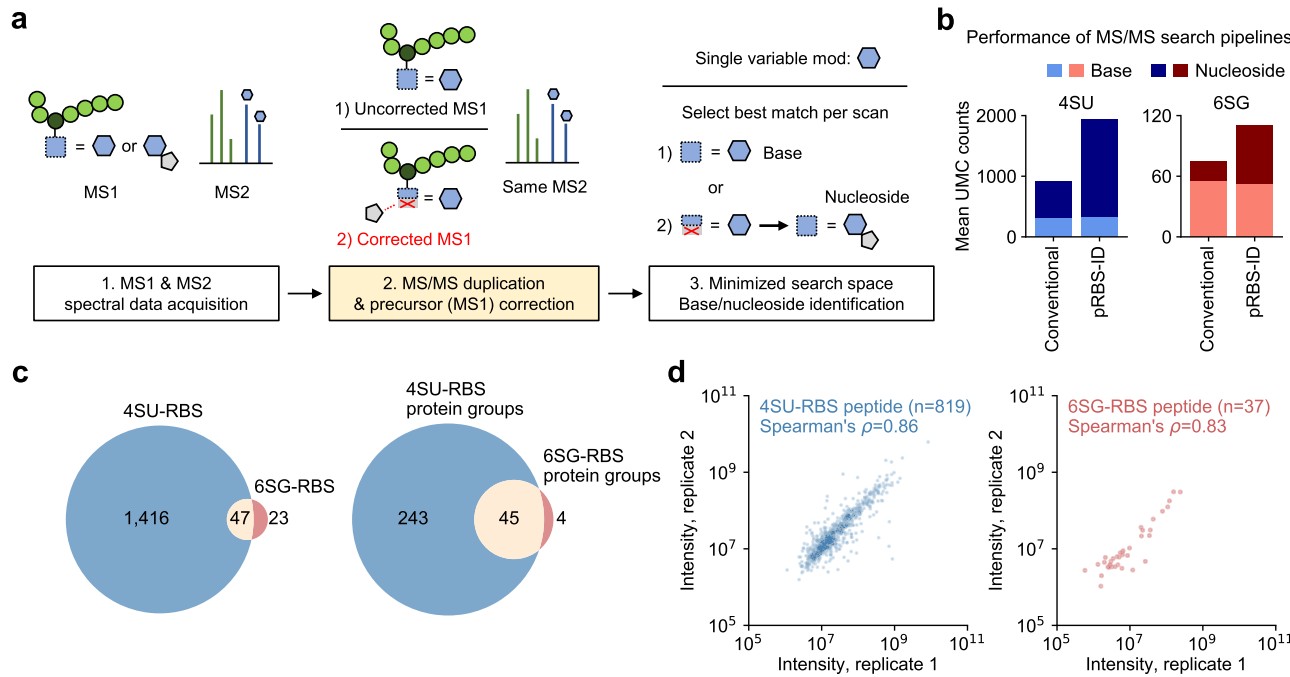

**Fig. 2 Robust PAR-RBS mapping by pRBS-ID. a** Schematic of the MS/MS search pipeline for pRBS-ID. Following MS/MS spectra acquisition, each MS2 scan is duplicated. One is left unaltered ("Uncorrected MS1"), while the other's precursor mass is corrected to account for the ribose neutral loss ("Corrected MS1"). The spectra are together searched against a target-decoy database, with a single variable modification of the base. Finally, a single best match is chosen for each scan. This allows for the simultaneous identification of peptides with base or nucleoside adducts. **b** Comparison of unique mass class (UMC) identification counts between conventional MS/MS search and pRBS-ID. In the former, variable modifications were defined as either base or nucleoside, without precursor correction considering ribose loss. **c** Comparison of identified 4SU-RBSs and 6SG-RBSs (left) or the respective protein groups (right). **d** MS1 intensity-based label-free quantification[24] of peptides containing 4SU-RBS (left) or 6SG-RBS (right) co-identified in replicate experiments. Spearman's correlation coefficients were calculated and rounded up to the second decimal point.

spectral data were matched to peptides (in other words, PSMs were generated) in a target-decoy database with a single variable modification of the PAR-base. For each MS/MS scan, a single best PSM was selected within the entire set of uncorrected and corrected scans, thereby excluding any chance of PSM inflation; just one PSM per a single MS/MS scan. If the best match came from the uncorrected scan, the MS1 modification would be the PAR-base form; for the corrected scan, the PAR-nucleoside form (Fig. 2a, step 3). In this way, both PAR-base or nucleoside form modifications could be identified from a minimal MS/MS search space using a single base form modification. Notably, because both 4SU and 6SG-nucleosides lose the identical mass of ribose, the same approach could be applied to both crosslink types.

Through careful inspection of the MS/MS data, we noticed that MS1 chromatograms from the same peptide showed multiple peaks with distinct retention times (Supplementary Fig. 1a, b). These can originate from various RBS localizations or crosslink isoforms[15] (i.e., different atom pairs in amino acid-PAR are crosslinked). Thus, each XIC peak should be distinguished for precise RBS localization and quantification. Therefore we integrated mPE-MMR[24], a tool that marks individual XIC peaks as separate unique mass classes (UMCs) and assigns label-free quantification values, prior to the MS-GF+[25] closed search in pRBS-ID (see online "Methods" for details). Notably, the assignment of one RBS localization per UMC allowed precise identification of multiple RBSs from a single peptide. This expanded the RBS repertoire compared to the previous analysis pipeline in RBS-ID, where only a single RBS was chosen for each peptide[15].

Finally, we evaluated the performance of the pRBS-ID MS/MS search pipeline against a conventional search method using the same dataset. For the "conventional" search, we analyzed PAR-base and nucleoside modifications independently, without MS1

correction or consideration of the ribose loss. pRBS-ID showed a remarkable increase in the PAR-nucleoside form identification while that of PAR-base form was comparable between the two methods (Fig. 2b). This was because pRBS-ID considers the transition of PAR-nucleoside forms in MS1 into PAR-base forms in MS2, enabling more sensitive identification compared to the conventional search.

**Robust and base-specific identification and quantification of PAR-RBS.** Using pRBS-ID, we identified 1,463 4SU-RBSs and 70 6SG-RBSs from 288 and 49 proteins, respectively (Fig. 2c, Supplementary Fig. 5, and Supplementary Data 2). As expected, the sites were found mainly in previously annotated RBPs[26] (Supplementary Fig. 6a, b). The number of 6SG-RBSs was smaller than that of 4SU-RBSs, although we used twice the amount of the sample, which is likely due to the relatively low UV-crosslinking efficiency of 6SG[17]. Twenty-three RBSs were found specific to 6SG (Fig. 2c). Other 6SG-RBSs crosslinked to both 4SU and 6SG, suggesting that these RBSs may be in close contact with adjacent U/G sequences or that the interactions may not be sequence-specific.

Moreover, pRBS-ID enabled accurate MS1 intensity-based label-free quantification[24] of RBSs. While pRBS-ID showed modest overlap between replicate experiments in terms of identification (Supplementary Fig. 5), the MS1 intensities of RBS-containing peptides co-identified in the replicate experiments showed high quantitative reproducibility (Fig. 2d).

**Comparison of base-specific features of RBSs.** Having generated the 4SU-RBS dataset by pRBS-ID, we first examined the differences in RBSs crosslinked to the same base type (U), but using different UV irradiation (UVA for 4SU; UVC for natural U).

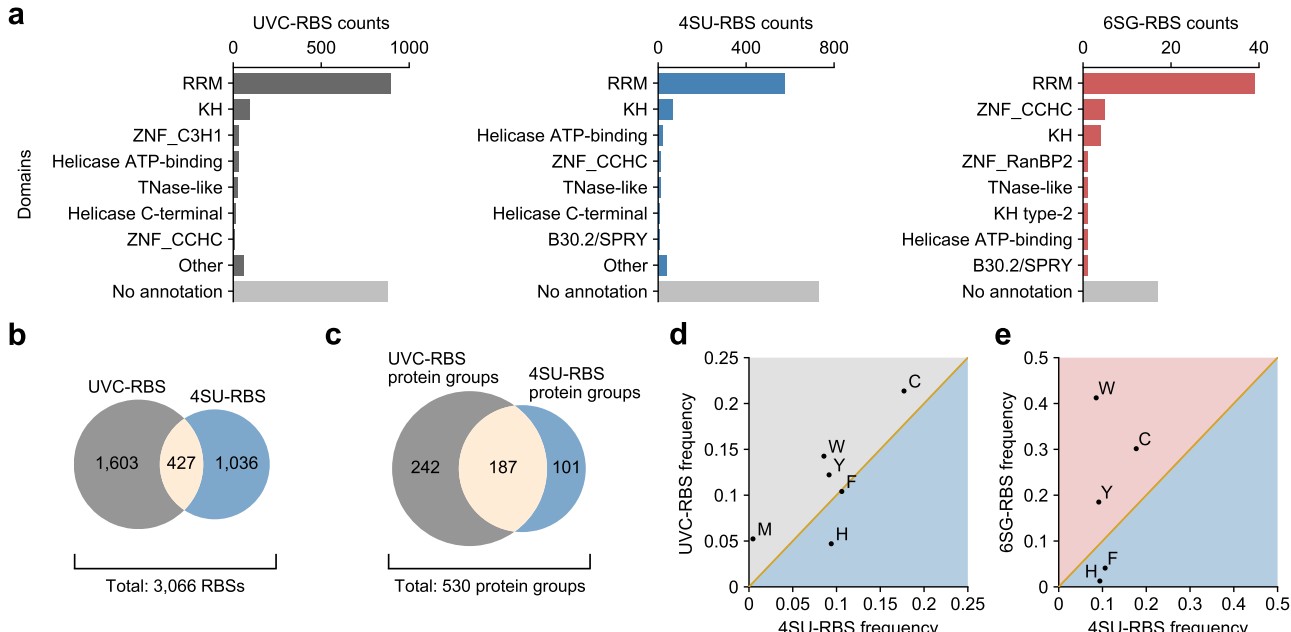

**Fig. 3 Comparison of base-specific RBS datasets. a** Domain annotation[26] of identified UVC-RBSs (left), 4SU-RBSs (middle), and 6SG-RBSs (right). Comparison of identified 4SU-RBSs and UVC-RBSs (**b**) or the respective protein groups (**c**). Comparison of amino acid frequencies between 4SU-RBS and UVC-RBS (**d**) or 6SG-RBS (**e**), respectively normalized by those from all sequences in proteins where each RBS type was identified. Amino acids that ranked in the top five are shown.

To obtain a comparable UVC-RBS dataset, we re-analyzed our previously published oligo d(T)-enriched mRNA-RBS dataset[15] by integrating mPE-MMR[24] into the MS/MS search pipeline in RBS-ID (see online methods for details). As a result, from the previous RBS-ID data (generated from 2.5 times more cells than the pRBS-ID experiment), 2,030 RBSs were identified (Supplementary Data 2). These RBSs were from 429 protein groups that are annotated mostly as RBPs (Supplementary Fig. 6c) and located mainly in canonical RNA-binding domains (RBDs) (Fig. 3a). Out of 2,030 RBSs identified by RBS-ID, only 427 sites were commonly detected in both RBS-ID and pRBS-ID experiments (Fig. 3b). pRBS-ID revealed 1,036 additional sites, which substantially expanded the list of U-contacting RBS to 3,066 sites in total. Thus, these two methods can be used to complement each other.

Of note, in the protein level, the RBS-ID and pRBS-ID have a larger overlap than they do in the RBS level (Fig. 3c). In terms of protein domains, the two datasets were highly comparable (Fig. 3a), including the identification of RNA-binding regions without domain annotation in canonical RBPs (Supplementary Fig. 7). These observations suggest that 4SU and U would crosslink to similar proteins and domains but different amino acids in the vicinity, likely by the virtue of their distinct preferences for UV-crosslinking with different amino acid types.

Hence, we compared the amino acid type frequencies between the 4SU- and UVC-RBSs (Fig. 3d). RNA base-interactors (Trp, Tyr, Phe; aromatic amino acids) and an efficient electron acceptor (Cys; thiol-containing amino acid) were the top four amino acid types in common, all showing modest enrichment as UVC-RBS except for Phe. Meanwhile, Met (thioester-containing amino acid) showed a high frequency only as UVC-RBS. This finding suggests that natural U crosslinks readily with any sulfur-containing amino acid (Cys, Met), while 4SU crosslinks only with one containing a thiol group (Cys). On the other hand, 4SU exhibited a broader preference for His, Arg, Lys (basic amino acids), and Pro, compared with natural U (Supplementary Fig. 8).

Thus, 4SU would be useful to broaden the coverage of RBS mapping. Overall, the overlapping yet distinct crosslinking preferences of the two UV-crosslinking methods indicate that they can complement one another to profile RBSs of different amino acid types, thereby effectively expanding the repertoire of RBSs.

Next, we analyzed the differences in RBSs crosslinked to distinct base types (U or G) using the 4SU-RBS and 6SG-RBS datasets. Although the number of identified 6SG-RBSs may be rather small to make highly accurate comparisons, the analysis can serve as a proof-of-principle to dissect the base-specific features of RNA−protein interactions. First, we compared the domain-level characteristics of these two types of PAR-RBSs[26] (Fig. 3a). Both RBS types were mainly identified within annotated RBDs, with the RRM domain as the top structural motif. The same was true for 6SG-RBSs, but notably, two zinc finger motifs, ZNF_CCHC and ZNF_RanBP2, were found more frequently with 6SG-RBSs than with 4SU-RBSs. ZNF_CCHC is known to bind G-rich RNA, with a notable example being LIN28 binding to the GGAG motif[27]. ZNF_RanBP2 binds to RNA with a GGU core motif[28], as exemplified in the interaction between FUS and the hnRNP A2/B1 pre-mRNA[29]. This indicates that pRBS-ID using 6SG successfully captures the G base-protein interactions in the cell.

In addition to the domain-level analysis, we further dissected the frequency of amino acid types in the two PAR-RBS datasets (Fig. 3e). Both PARs shared Trp, Tyr, Phe, His, and Cys as the top five most frequently captured amino acids. They were either RNA base-interactors (Trp, Tyr, Phe, His; aromatic ring- or π-bond-containing amino acids) or an efficient electron acceptor (Cys; thiol-containing amino acid) that can facilitate efficient UV-crosslinking. Among the five amino acids, Trp, Cys, and Tyr showed higher frequencies as 6SG-RBS, while His and Phe were preferred as 4SU-RBS. We predict that the preferences may result from a combined effect of the prevalence of a base type at RNA-protein interfaces and differential UV-crosslinking efficiency for each amino acid type.

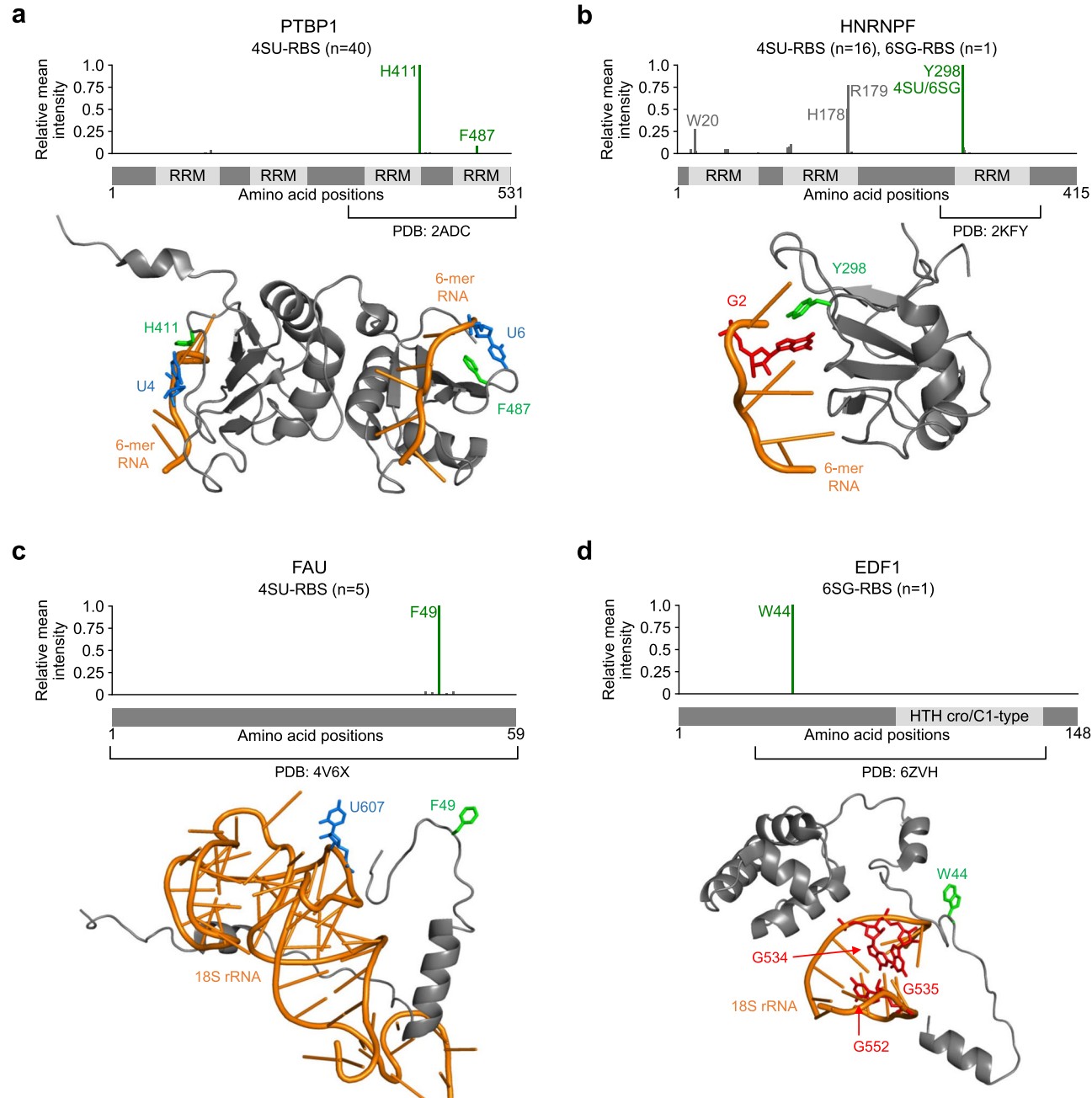

**Fig. 4 Base-specific RNA−protein interactions in RNP structures. a−d** RBSs identified in this study (top) and their positions in known structures (bottom). One or two top-ranking RBSs from each protein are highlighted in green. The U and G residues proximal to RBSs are shown in blue and red, respectively. **a** Interaction between H411 or F487 in PTBP1 (4SU-RBSs) with proximal Us in structural data (PDB: 2ADC[30]). **b** Interaction between Y298 in HNRNPF (both 4SU- and 6SG-RBS) with proximal G in structural data (PDB: 2KFY[31]). **c** Potential interaction between F49 in FAU (4SU-RBS) and most proximal U in structural data (PDB: 4V6X[33]). **d** Potential interaction between W44 in EDF1 (6SG-RBS) and proximal Gs in structural data (PDB: 6ZVH[34]).

**Structural insights into RNA−protein interactions using pRBS-ID.** pRBS-ID can pinpoint the exact amino acids that interact with RNA at "zero distance". We revisited previous high-resolution structural datasets on RNA-protein complexes and checked whether 4SU- or 6SG-RBSs interact with U or G bases, respectively. In PTBP1, two prominent 4SU-RBSs (H411 and F487) are located close to the U bases[30] (Fig. 4a). Furthermore, in hnRNP F protein, Y298 was prominently identified as both 4SU- and 6SG-RBS (Fig. 4b). Y298 is adjacent to the G base in the structure[31]. Although no close interactions with a U residue could be found due to the particular RNA sequence they used for structural studies, our data suggest that,

in cells, hnRNP F Y298 may interact with both U and G in diverse pre-mRNA partners.

Structural approaches are instrumental in understanding the molecular basis of RNA−protein interactions[32], yet the resolved structures are static snapshots. Hence, dynamic intermolecular binding events, particularly those involving flexible loops and intrinsically disordered regions (IDRs), may not be captured in the conformation resolved by X-ray crystallography or cryo-EM. In theory, pRBS-ID can identify RBSs in RNP complexes at any conformational state. Thus, we examined RBSs on flexible regions to assess our hypothesis. In FAU, a component of the human 40S ribosome, F49 was identified as 4SU-RBS[33] (Fig. 4c). According

to the cryo-EM structure, this RBS does not directly interact with rRNA. However, F49, as part of a disordered region, may be brought closer to U607 in the 18S rRNA for direct interaction. Another example was W44 (6SG-RBS) in EDF1, which binds to collided ribosomes to mediate the ribosome-associated protein quality control pathway[34] (Fig. 4d). W44 of EDF1 is located within a long disordered region and may contact G534, G535, or G552 in the 18S rRNA, which could not be captured in the cryo-EM structure. Taken together, these results suggest that pRBS-ID datasets can be valuable resources for studies of RNP complexes to elucidate the RNA−protein interactions in both stable and dynamic states.

## Discussion

In this study, we developed pRBS-ID, a method that robustly profiles PAR-crosslinked RBSs in proteins. The benefits of PAR-crosslinking[17–21] and chemical RNA cleavage[15] are combined in this method. PAR-crosslinking provides selectivity for labeled RNA and expands the RBS repertoire from those crosslinked to natural U by UVC to those readily crosslinked by UVA to 4SU and 6SG. Meanwhile, the chemical cleavage of RNA using HF minimizes the complexity of RNA adduct moieties to allow precise RBS identification.

In terms of the data analysis method, we present a new search pipeline optimized for robust PAR-RBS identification, based on the PAR adducts' chemical features, which we discovered in this study. The search pipeline enables accurate MS1-level quantification of RBS-peptides, which can be utilized for single amino acid-level differential analysis of RNA−protein interactions in distinct biological conditions.

Using pRBS-ID, we systematically profiled 1,463 4SU-RBSs along with 70 6SG-RBSs. Adding onto the RBSs crosslinked to natural U bases, the PAR-RBSs provided a substantially expanded landscape of RBSs in the human proteome, cataloging a total of 3,077 RBSs in 532 protein groups (Supplementary Fig. 9). If photoactivatable adenosine and cytosine analogs are developed in the future, pRBS-ID will be able to cover all base types.

pRBS-ID can be used as a widely applicable and amenable platform for investigating nucleic acid−protein interactions. When combined with pulse labeling and specific RNA probes, the method can profile RBSs in proteins bound to specific RNA types at various temporal stages. For instance, RBSs in proteins bound to viral genomic RNA or viral subgenomic RNAs can be selectively captured through metabolic labeling at a particular stage of the viral life cycle. In addition, RBPs acting at different stages of RNA lifetime can be distinctively captured via pRBS-ID. Importantly, the PAR-adducts provide direct evidence for RNA binding, thereby offering strong confidence in the identification. Furthermore, the single amino acid-resolution of RBS mapping can be of great value in unraveling the action mechanisms of RBPs. On the other hand, the analytical procedure of PAR adduct characterization using combinations of open search tools can be benchmarked to characterize other nucleic acid adducts. These include different nucleic acid types such as DNA[35], xeno nucleic acid (XNA), or chemical crosslinking reagents such as formaldehyde[36,37]. A MS/MS search pipeline can then be optimized for each crosslink type, as in the pRBS-ID method development.

In addition to the utility as a methodological platform, the pRBS-ID dataset, together with the previously published RBS-ID dataset, can serve as a valuable resource for the structural prediction of RNA−protein interactions[32]. As exemplified in Fig. 4, RBSs can be mapped into individual RBP structural data of interest to interpret the RNA-binding conformation. A longer-shot approach would be related to the recent 'game-changing'

progress in de novo single protein structure prediction[38–40], resulting in AlphaFold[41] and RoseTTAFold[42]. If this approach is extended to the de novo structural determination of RNA-protein complexes, the RBS datasets can be integrated to enhance the accuracy and extent of RNP conformational prediction. pRBS-ID together with structural and biochemical methods will also be valuable for the development of small molecule-based therapeutics that interfere with RNA−protein interaction[43] as a number of disease-associated RNAs and their protein partners are considered promising as drug targets. This integrative approach will be particularly useful for investigating the RNA-binding modes of RBPs with flexible loops and IDRs that are difficult to solve the structure. Proteins with IDRs play critical roles in complex RNA−protein interaction and condensate formation in normal development and neuronal pathology. Overall, pRBS-ID can work as a powerful, general, and amenable method to deepen our knowledge of the dynamic interactions between proteins and nucleic acids.

Limitations of the current study include the low coverage of 6SG-RBSs owing to its inefficient crosslinking efficiency. This can be improved by increasing the input amount or using enhanced separation technologies prior to the MS/MS analysis. Thus, for general purposes to identify and profile RBPs and RBSs, 4SU would be more useful than 6SG. However, for RBPs that interact specifically with G residues, 6SG would be a highly valuable tool, especially when applying this method to purified or enriched proteins of interest. In addition, we would like to call attention to the strong toxicity of HF which should be handled with great caution and inside a fume hood[44]. Finding safer alternative chemicals with similar hydrolyzing activity would greatly help resolve the issue and broaden the usage of chemical RNA digestion to map RBSs.

## Methods

**Cell culture**. HeLa cells used in this study were 2D-cultured in Dulbecco's modified Eagle's medium (Welgene) supplemented with 9% FBS (Welgene) and 1× antibiotic-antimycotic (Gibco), in a 37 °C incubator with a $CO_2$ concentration of 5%. HeLa cells were tested negative for mycoplasma contamination, and authenticated by ATCC by short tandem repeat profiling following ISO 9001:2008 and ISO/IEC 17025:2005 quality standards. HeLa cells were generous gifts from laboratories in the School of Biological Sciences, Seoul National University.

**pRBS-ID sample preparation and LC-MS/MS**. *PAR labeling and UVA-crosslinking* − HeLa cells were grown on 150 Φ dishes (SPL) to 40−50% (4-thiouridine) or 70−80% (6-thioguanosine) confluency. The cells were labeled twice with media containing 100 μM 4-thiouridine (4SU, Merck) or 6-thioguanosine (6SG, Ambeed, Inc.) for 12 h each. After incubation, the cells were washed twice with PBS (Amresco), then placed on ice in UVP 95-0228-01 Longwave Ultraviolet Crosslinker (Analytik Jena US), 12 cm away from the UV lamp. The cells were irradiated with 450 mJ of 365 nm UV.

*Oligo-d(T) enrichment and tryptic digestion* − UVA-crosslinked HeLa cells on each 150 Φ dish were lysed on-dish using 1.2 ml lysis buffer (20 mM Tris pH 7.5, 500 mM NaCl (Merck), 1 mM EDTA (Ambion), 0.5% SDS (Merck)) supplemented with 18 μl of SUPERase•In™ RNase Inhibitor (Invitrogen). The lysates from eight dishes (twice the amount required for LC-MS/MS) were merged, homogenized with a 21 G syringe, added to 4 ml oligo-dT(25) beads (New England Biolabs) pre-equilibrated with lysis buffer, and incubated for 1 h at RT. The supernatant was removed, and the beads were washed once with 1 ml lysis buffer, then once more using 1 ml wash buffer (20 mM Tris pH 7.5, 500 mM NaCl, 1 mM EDTA). The beads were moved to a new tube at the initial wash with lysis buffer. Following washes, 1 ml of digest buffer (50 mM Tris pH 8.0 and 5 mM EDTA) was added. 1,4-dithiothreitol (Merck) was added to each sample to 10 mM and incubated for 1 h at 37 °C, and iodoacetamide (IAA, Merck) was added to 40 mM and incubated for 1 h at 37 °C. Then, 15 μg trypsin (MS grade, Thermo Fisher Scientific) was added and incubated overnight at 37 °C. The samples were eluted by heating at 95 °C for 5 min.

*RNA-peptide conjugate enrichment by size-cutoff filtration* − A 0.5 ml Amicon Ultra 30 kDa Centrifugal Filter Unit (Merck) was equilibrated twice using 0.5 ml Equilibration buffer (10 mM Tris pH 8.0). The centrifugation steps using the Amicon filters occurred at 14,000 × g for 15 min. The eluted sample was passed through the filter and topped seven times more with a 0.5 ml Equilibration buffer. This allowed the removal of free peptides and the retention of RNA-crosslinked

peptides on the filter. Another 0.5 ml Equilibration buffer was added to collect the RNA-peptide conjugates by inverting the filter and spinning briefly. This step was repeated once more.

*HF digestion and peptide clean-up* − The sample volume was adjusted to 100 μl using Concentrator plus system (Eppendorf) set to RT. Then, 400 μl of 48% HF was added and locked into a CaCO₃ trap. The sample was incubated at 4 °C overnight and dried in a Concentrator plus system supplemented with a CaCO₃ trap at RT. The sample was reconstituted with triple distilled water and cleaned up using ZipTip (Millipore). The eluted peptides were dried in the Concentrator plus at RT, and reconstituted in 50 mM ABC, pH 8.5. Following centrifugation at $13,000 \times g$ for 5 min at RT, the supernatant was collected, and a half was injected for LC-MS/MS. All experimental steps exposing HF to air were carried out inside a fume hood with great caution. Calgonate® gels (Calgonate) were kept readily available for emergency use, and all procedures followed the safety measures provided by the manufacturer.

*LC-MS/MS for pRBS-ID* − LC-MS/MS analysis was performed using an Orbitrap Fusion Lumos Tribrid MS (Thermo Fisher Scientific) coupled with a nanoAcquity system (Waters) equipped with a capillary analytical column (75 μm i.d. × 100 cm) and trap column (150 μm i.d. × 3 cm), both in-house packed with 3 μm Jupiter C18 particles (Phenomenex). The LC flow rate was 300 nl min⁻¹ with a 100 min linear gradient ranging from 95% solvent A (0.1% formic acid (Merck)) to 40% solvent B (100% acetonitrile, 0.1% formic acid). Full MS scans (m/z 300–1,500) were acquired at a resolution of 60k (at m/z 200). Higher-energy collisional dissociation (HCD) fragmentation was performed under 30% of normalized collision energy (NCE) via precursor isolation within 1.4 Th window. The MS2 scans were acquired at a resolution of 15k, with a maximum precursor ion injection time (ITmax) of 30 ms and automatic gain control (AGC) of $1 \times 10^4$.

**Open search and MS2 modification search.** *Open search* −MODa (v1.6.0)[23] search was performed on modification sizes 0–400, using mzXML files generated from RawConverter (v.1.1.0.19)[45], in a data-dependent acquisition of charge states 2–7. The target-decoy database was generated by MS-GF+ (v.20170127)[25] from the Swiss-Prot human database (May 2019)[26]. The PSM-level false discovery rate (FDR) was set to <1% for 4SU, or <5% for 6SG. We analyzed the frequently modified mass using moda.ptm files. For modifications on Cys, the modified mass was adjusted by adding 57 to account for fixed modification (carbamidomethylation). An MSFragger (v.20190530)[22] open search via Fragpipe was performed with raw files on modification sizes 0–400. The target-decoy database and the FDR of the PSMs sorted by hyperscore were identical to those of MODa search. We took massdiff values in the tsv files to analyze the frequently modified mass. Adjustment of modification on Cys was not performed, as MSFragger open search cannot specify the modified site within peptides.

*MS2 modification search* − PSMs with rounded massdiff values of 94 (4SU-base), 226 (4SU-nucleoside), 133 (6SG-base), 265 (6SG-nucleoside) were collected from the FDR-filtered MSFragger search results, respectively. Then, y-ions harboring all potential 4SU or 6SG modification sites in the best_locs column (amino acids in lowercase) were collected for each PSM. Nominal modification masses ranging from 1 to 400 were iteratively given to the y-ion masses and matched against the MS2 peaks of corresponding scans from MXFiltered.mgf file generated by msconvert (v.3.0.1908)[46] integrated into mPE-MMR (v.1.1.8)[24]. The relative MS2 peak intensities of matched ions (0.5 Da window) were binned for each modification mass. The summed relative intensities were normalized against the maximum value.

*Spectrum visualization* − Thermo Xcalibur Qual Browser (Thermo Fisher Scientific) and its boxcar method for peak smoothing were used to visualize extracted ion chromatogram (XIC) of precursor ions. Individual MS2 spectra annotations (0.5 Da matching window) were based on peaks from MXFiltered.mgf files and visualized by custom python codes.

**Closed search and data processing.** *pRBS-ID closed search with precursor mass correction* − Raw files were processed with mPE-MMR[24] using default settings. Followingly, each scan in the MXFiltered.mgf file was duplicated with a precursor mass correction of ribose ($C_5H_8O_4$) removal. MS-GF+ search[25] was performed with the corrected mgf file. Fixed modification considered was $C_2H_3N_1O_1$ (carbamidomethylation) on Cys at any residue. Variable modifications considered were $C_2H_{-1}N_1$ (4SU-base minus $H_2S$ minus carbamidomethylation) on Cys and $C_4H_2N_2O_1$ (4SU-base minus $H_2S$) on the remaining 19 amino acids for 4SU datasets; $C_3N_4O_{-1}$ (6SG-base minus $H_2S$ minus carbamidomethylation) on Cys and $C_5H_3N_5$ (6SG-base minus $H_2S$) on remaining 19 amino acids, for 6SG datasets. Modification on Cys residues was altered to distinguish base-crosslinked Cys from the carbamidomethylation of uncrosslinked free Cys residues by IAA. Only one modification was allowed per peptide. The Swiss-prot human database[26] (May 2019) was used, and the decoy database was generated by MS-GF+[25]. The output mzid file was converted to a tsv file for post-processing.

*Conventional closed search without precursor mass correction* − Raw files were processed with mPE-MMR[24], and MS-GF+ search[25] was performed with the MXFiltered.mgf file. Fixed modification considered was $C_2H_3N_1O_1$ (carbamidomethylation) on Cys at any residue. Variable modifications considered were $C_7H_9N_1O_5$ (uridine minus carbamidomethylation) on Cys and $C_9H_{12}N_2O_6$ (uridine) for U mRNA-RBS datasets[15]; $C_2H_{-1}N_1$ (4SU-base minus $H_2S$ minus

carbamidomethylation) or $C_7H_7N_1O_4$ (4SU-nucleoside minus $H_2S$ minus carbamidomethylation) on Cys and $C_4H_2N_2O_1$ (4SU-base minus $H_2S$) or $C_9H_{10}N_2O_5$ (4SU-nucleoside minus $H_2S$) on remaining 19 amino acids, for 4SU datasets; $C_3N_4O_{-1}$ (6SG-base minus $H_2S$ minus carbamidomethylation) or $C_8H_8N_4O_3$ (6SG-nucleoside minus $H_2S$ minus carbamidomethylation) on Cys and $C_5H_3N_5$ (6SG-base minus $H_2S$) or $C_{10}H_{11}N_5O_4$ (6SG-nucleoside minus $H_2S$) on remaining 19 amino acids, for 6SG datasets. The rest were identical to *pRBS-ID closed search with precursor mass correction*.

*Further-processing for robust RBS localization and* quantification − For each experiment, PSMs within the ±5 ppm window from the mean precursor isotopic error of highly scored PSMs (e.g., MS-GF + Q-Value = 0) were collected. Then, PSMs with the lowest SpecEValue from MS-GF+ were taken per scan (for closed search with precursor mass correction, single PSM from MS1 corrected + uncorrected PSM pool). Unmodified PSMs and those without unique RBS localization were discarded to remove ambiguity (including those localized to the C-terminal Arg/Lys of peptides). Precursor modifications of each PSMs were annotated as the base (for closed search with precursor mass correction, uncorrected; for conventional closed search, variable modifications corresponding to bases) or nucleoside (for closed search with precursor mass correction, corrected; for conventional closed search, variable modifications corresponding to nucleosides). Then, UMCs were assigned to each PSM using UMC_MSMS_MX.txt file from mPE-MMR[24]. For each UMC, RBS localization scores were calculated as (sum of -$\log_{10}$(SpecEValue) of RBS PSMs in UMC corresponding to the individual localization) divided by (sum of -$\log_{10}$(SpecEValue from ms-gf+) of all RBS PSMs in UMC). A single RBS localization with the highest localization score was assigned for each UMC. Then, a UMC-level FDR cutoff of <0.01 was applied. MS1 intensity-based label-free quantification value for each UMC was designated as the MostAbundant value in UMC.txt file from mPE-MMR. RBS locations within proteins were mapped to the proteome databases used above. Finally, the UMC identification and label-free quantification results were merged into RBS-level or RBS-peptide-level. For each crosslink type (UVC, 4SU, or 6SG), amino acid frequencies of RBSs were calculated as (RBS counts of each amino acid) divided by (counts of each amino acid in the entire sequence of proteins with RBS) and normalized such that the sum equals 1.

The MS proteomics data have been deposited to the ProteomeXchange Consortium via the PRIDE partner repository[47] with the dataset identifier PXD023401[48]. The custom Python codes central to this study have been deposited to the GitHub repository[49].

*Integration of previous studies and databases* − Identified RBSs were mapped to RBP annotations in UniProtKB (June 2019), domain annotations in UniProt PROSITE (June 2019), UniProt zinc finger and coiled-coil annotations (August 2019) if unmapped to UniProt PROSITE[26]. Structural data were obtained from Protein Data Bank with the PDB codes 2ADC[50] (PTBP1), 2KFY[51] (HNRNPF), 4V6X[52] (FAU), and 6ZVH[53] (EDF1), which were visualized using PyMOL (v.1.7.2.1.).

**Reporting summary.** Further information on research design is available in the Nature Research Reporting Summary linked to this article.

## Data availability

The mass spectrometry proteomics data generated in this study have been deposited in the ProteomeXchange Consortium PRIDE database under accession code PXD023401. The structural data used in this study are available in the RCSB PDB database under accession codes 2ADC, 2KFY, 4V6X, and 6ZVH. The raw data were used to generate Figs. 1–4. Source data are provided with this paper.

## Code availability

The custom Python codes central to this study have been deposited to the GitHub repository.

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

## Acknowledgements

We sincerely thank Pacific Northwest National Laboratory and the OMICS.PNL.GOV for providing valuable software. We are also grateful to S. Shin, Y. Na and B. Um for insightful comments and discussion, J. Kim, J. Yang, D. Choi, E. Kim and E. Chang for technical help, and all members of our laboratories for helpful discussion. This work was supported by IBS-R008-D1 of Institute for Basic Science from the Ministry of Science and ICT of Korea (J.W.B., V.N.K. and J.-S.K.), grant from National Research Foundation of Korea (NRF-2021R1A2C2009336, J.-S.K.) and the BK21 Research Fellowships (J.W.B.) from the Ministry of Education, Science and Technology of Korea.

## Author contributions

J.W.B., V.N.K. and J.-S.K. conceived the project and designed the experiments; J.W.B. developed the protocol and performed all biochemical and LC-MS/MS experiments; J.W.B. and S.K. analyzed all LC-MS/MS datasets; J.W.B., V.N.K. and J.-S.K. wrote the paper.

## Competing interests

S. K. has a financial interest in Seer Inc. The other authors declare no competing interests.
