## [Peer Review File · Nature Communications]

Photoactivatable ribonucleosides mark base-specific RNA-binding sitesReviewers' Comments:

Reviewer #1:

Remarks to the Author:

The manuscript by Bae and colleagues describes development and use of pRBS-ID to identify photo-crosslinked peptides with single amino acid-resolution and base specificity. The authors utilize the photo-reactive nucleoside analogs 4-thiouridine (4U) and 6-thioguanosine to increase crosslinking efficacy and apply their previously established RBS-ID protocol and work flow (Bae et al NSMB 2010). Bae and coworkers report uridine and guanosine crosslinked peptides in HeLa cells.

Overall, the work is very extensive and quite thorough, to say the least. Yet despite all the good work it does not take us that much further from the previously published approach RBS-ID, which was published as a technical report in NSMB in 2020. The authors suggest themselves that the RBS-ID and pRBS-ID approaches complement each other due to the discovery of different sets of crosslinked peptides. The authors find a larger number of crosslinked peptides but overall the number of protein groups using both approaches is very similar. The number of guanosine crosslinked peptide is only a minor fraction of the overall identified peptides and in its entirety not very insightful. The work was conducted as in the previous in HeLa cells for comparison but no other cellular system was examined. Furthermore, any discussion on newly discovered RNA-binding protein and/or protein domains is missing. In summary, the manuscript in its current form, due to the limited mechanistic and biological insights, does not warrant publication in Nature Communication but is more suited for specialized journals like RNA or RNA Biology.

Reviewer #2:

Remarks to the Author:

Comments for the Author

Summary

Bae and colleagues present a manuscript entitled "Photoactivatable ribonucleosides mark base-specific RNA-binding sites" which is an extension of earlier work from the same labs that used chemical removal of RNA adducts to proteins/peptides after UV cross-linking and final identification of RNA-binding sites (RBSs) in proteins. In the present study, the authors combine this with UV irradiation (365 nm) of photoactivatable nucleotides and establish a modified mass spectrometry analysis pipeline to look at the data from what is now called pRBS-ID.

General remarks

The manuscript is well written and understandable. The images are clear and the particularly the schematics help to understand the experimental approach.

Overall, the manuscript is interesting and a logical next step of the recently-published HF-based RNA cleavage protocol which was done using 254 nm UV cross-linking. The method thus extends the available repertoire of techniques to identify RBSs in proteins. The study focuses on the method itself and the presented data is chosen to validate the technique with already-known RBPs/RBSs instead of presenting novel and/or unexpected sites.

Major criticism

My biggest concern is the presentation of the method in comparison to other protocols and how the authors justify their "p" in pRBS-ID.

- First of all, the manuscript states that UVC (254 nm) cross-linking results in uridines (U) being cross-linked exclusively. While it is true that U is found as the nucleotide that cross-links most efficiently, the cited studies (as well as other literature) do not support exclusive cross-linking with U.
- as a result of this, the reasoning of why we need a new method that goes beyond "U cross-linking" is

not really a good reason d'être for pRBS-ID

- to make things worse, the authors then use 4SU-based cross-linking at 365 nm which truly is an exclusive U cross-link (only 4SU-modified nucleotides can form cross-links) but they call the 254 nm cross-link "U-RBS" (Fig. 3). This is not only confusing but misleading in my opinion and I strongly recommend to reconsider the labeling of the UVA samples.

- Another issue is how the authors describe work of others to claim that their new approach is necessary. While I agree that HF-based removal of RNA is superior to enzymatic cleavage for the described reasons, the authors make unfair comparisons with older methods:

- l. 72ff: The authors suggest that RNPxl and TRAPP have only had "limited success" in identifying RBSs. A rather bold statement for a study that finds 70 sites using the 6SG nucleotide (which is in the ballpark of the criticised papers).

- l. 76 ff: The authors claim that their approach is much more efficient in terms of sample size while comparing HeLa cell culture to yeast. This is like comparing apples and pies as yeast as a liquid (and turbid) culture is way harder to cross-link than a monolayer cell line in a dish. Thus, the large input amount for yeast is most likely a result of poor UV cross-linking than of the downstream approach. I recommend to find a better comparison, or to support your claims by providing experimental data of pRBS-ID from yeast.

- The large difference between U and G sites (after 4SU or 6SG incorporation, respectively) is worrisome, particularly since the authors describe the need to have a method that does not only focus on U (see comment above). I agree however that this is a useful feature and, as proof-of-principle, showing the potential of pRBS-ID. However, one concern is how much statistical data can really be derived from such a rather small dataset

- Fig. 3 c/d only compares 254 with 4SU. It would still be interesting if the "G cross-links" are a subset of the UVA-found RBSs or rather novel.

- For Fig. 3f, my question is how statistically sound the analysis can be for 70 sites and 20 (potential) amino acids. Can you provide the frequencies of the other 15 (if only for this review and not as part of the paper) of 3f and 3g?

- Finally, I am missing a discussion of the limitations of the method. Hydrogen fluoride is not at all easy to handle (toxicity) and not each lab in the field has been able to take up the approach due to this. I am sure that the authors can point out other potential issues that help the reader to make an informed call when pRBS-ID is the best method to pick up and when other protocols are superior.

Minor points

- Fig 4: The plots lack y-axis markings. Despite the values being artificial units, it would be great to have some graphical cues to be able to better compare the intensities of different green/grey bars in one protein.

- Results section from line 99 on: Since Nature Communications is not a mass spec-centric journal, I recommend to spend one or two sentences to quickly explain what the difference between M1 and M2 spectra is. This will render the manuscript much more approachable to a large audience to be able to follow the reasoning of the newly-established analysis pipeline.

- Methods: The link to the Github page which should contain the code/scripts for data analysis does not work (yet?).

Response to Reviewers

We thank the reviewers for carefully reading our manuscript and providing constructive feedback. We addressed the concerns and questions raised by the reviewers. We hope that the reviewers find our responses clear and to the point. Please note that the response will refer to the numberings as in the revised manuscript.

Reviewer #1 (Remarks to the Author):

The manuscript by Bae and colleagues describes development and use of pRBS-ID to identify photo-crosslinked peptides with single amino acid-resolution and base specificity. The authors utilize the photo-reactive nucleoside analogs 4-thiouridine (4U) and 6-thioguanosine to increase crosslinking efficacy and apply their previously established RBS-ID protocol and work flow (Bae et al NSMB 2010). Bae and coworkers report uridine and guanosine crosslinked peptides in HeLa cells.

Overall, the work is very extensive and quite thorough, to say the least. Yet despite all the good work it does not take us that much further from the previously published approach RBS-ID, which was published as a technical report in NSMB in 2020. The authors suggest themselves that the RBS-ID and pRBS-ID approaches complement each other due to the discovery of different sets of crosslinked peptides. (1) The authors find a larger number of crosslinked peptides but overall the number of protein groups using both approaches is very similar. (2) The number of guanosine crosslinked peptide is only a minor fraction of the overall identified peptides and in its entirety not very insightful. (3) The work was conducted as in the previous in HeLa cells for comparison but no other cellular system was examined. (4) Furthermore, any discussion on newly discovered RNA-binding protein and/or protein domains is missing.

- We thank the reviewer for the valuable feedback.

(1) Complementarity of RBS-ID and pRBS-ID: It is true that two methods (RBS-ID and pRBS-ID) capture partially overlapping protein pools. But please understand that our main purpose is to identify RBSs rather than to discover new RBPs. In the current study, pRBS-ID adds on to the RBPs identified by RBS-ID (~23% increase in the number of protein groups). More importantly, pRBS-ID expands the RBS catalogue substantially (~51% increase in the number of RBSs). So we believe that pRBS-ID is a truly complementary method to the existing RBS-ID, providing a valuable resource for studying the functions of individual RBPs.

Fig. 3b-c

Fig. 3b-c. Comparison of identified 4SU-RBSs and UVC-RBSs (b) or the respective protein groups (c).

2) The number of guanosine crosslinked peptides: We acknowledge in the manuscript the low coverage of 6SG-RBSs that owes to the inefficient crosslinking properties of 6SG. Nevertheless, we'd like to emphasize that this work is the first report of G-crosslinked RBSs. Even with the current sensitivity, 6SG will be useful for *in vitro* approaches to find RNA-binding residues in G-rich RNA-binding proteins. In the future, our pRBS-ID method can provide broader coverage for 6SG if combined with advanced LC-MS techniques such as trapped ion mobility MS platform (timsTOF SCP) or true nanoelectrospray technology. We discuss this point in lines 346-355, as follows.

Lines 346-355:

“Limitations of the current study include the low coverage of 6SG-RBSs owing to its inefficient crosslinking efficiency. This can be improved by increasing the input amount or using enhanced separation technologies prior to the MS/MS analysis. Thus, for general purposes to identify and profile RBPs and RBSs, 4SU would be more useful than 6SG. However, for RBPs that interact specifically with G residues, 6SG would be a highly valuable tool, especially when applying this method to purified or enriched proteins of interest.”

3) Application on other cellular systems: Given that our work is rather method-oriented, we believe that experiments using HeLa cells yielded sufficient data to develop the pRBS-ID pipeline. We discuss in detail the potential uses of pRBS-ID for dissecting RBP functions in diverse biological contexts in lines 313-327.

Lines 313-327:

“pRBS-ID can be used as a widely applicable and amenable platform for investigating nucleic acid-protein interactions. When combined with pulse labeling and specific RNA probes, the method can profile RBSs in proteins bound to specific RNA types at various temporal stages. For instance, RBSs in proteins bound to viral genomic RNA or viral subgenomic RNAs can be selectively captured through metabolic labeling at a particular stage of the viral life cycle. In addition, RBPs acting at different stages of RNA lifetime can be distinctively captured via pRBS-ID. Importantly, the PAR-adducts provide direct evidence for RNA binding, thereby offering strong confidence in the identification. Furthermore, the single amino acid-resolution of RBS mapping can be of great value in unraveling the action mechanisms of RBPs.”

4) Discussion on newly discovered RNA-binding proteins/domains: We amended the text to describe the potential RNA-binding modes of the novel domains in a structural perspective in Fig. 4c-d. Also, we elaborate on the benefits of pRBS-ID in dissecting the emerging roles of intrinsically disordered regions (IDRs) in RNA-protein interactions in lines 278-293 and 328-345 of the Results and Discussion section, respectively.

Fig. 4c-d

Fig. 4c-d. Base-specific RNA-protein interactions in RNP structures.

RBSs identified in this study (top) and their positions in known structures (bottom). One or two top-ranking RBSs from each protein are highlighted in green. The U and G residues proximal to RBSs are shown in blue and red, respectively. **c** Potential interaction between F49 in FAU (4SU-RBS) and most proximal U in structural data (PDB: 4V6X³³). **d** Potential interaction between W44 in EDF1 (6SG-RBS) and proximal Gs in structural data (PDB: 6ZVH³⁴).

Lines 278-293:

“Structural approaches are instrumental in understanding the molecular basis of RNA-protein interactions³², yet the resolved structures are static snapshots. Hence, dynamic intermolecular binding events, particularly those involving flexible loops and intrinsically disordered regions (IDRs), may not be captured in the conformation resolved by X-ray crystallography or cryo-EM. In theory, pRBS-ID can identify RBSs in RNP complexes at any conformational state. Thus, we examined RBSs on flexible regions to assess our hypothesis. In FAU, a component of the human 40S ribosome, F49 was identified as 4SU-RBS³³ (**Fig. 4c**). According to the cryo-EM structure, this RBS does not directly interact with rRNA. However, F49, as part of a disordered region, may be brought closer to U607

in the 18S rRNA for direct interaction. Another example was W44 (6SG-RBS) in EDF1, which binds to collided ribosomes to mediate the ribosome-associated protein quality control pathway³⁴ (**Fig. 4d**). W44 of EDF1 is located within a long disordered region and may contact G534, G535, or G552 in the 18S rRNA, which could not be captured in the cryo-EM structure. Taken together, these results suggest that pRBS-ID datasets can be valuable resources for studies of RNP complexes to elucidate the RNA-protein interactions in both stable and dynamic states.”

Lines 328-345:

“In addition to the utility as a methodological platform, the pRBS-ID dataset, together with the previously published RBS-ID dataset, can serve as a valuable resource for the structural prediction of RNA-protein interactions³². As exemplified in Figure 4, RBSs can be mapped into individual RBP structural data of interest to interpret the RNA-binding conformation. A longer-shot approach would be related to the recent ‘game-changing’ progress in de novo single protein structure prediction³⁹⁻⁴¹, resulting in AlphaFold⁴² and RoseTTAFold⁴³. If this approach is extended to the de novo structural determination of RNA-protein complexes, the RBS datasets can be integrated to enhance the accuracy and extent of RNP conformational prediction. pRBS-ID together with structural and biochemical methods will also be valuable for the development of small molecule-based therapeutics that interfere with RNA-protein interaction⁴⁴ as a number of disease-associated RNAs and their protein partners are considered promising as drug targets. This integrative approach will be particularly useful for investigating the RNA-binding modes of RBPs with flexible loops and IDRs that are difficult to solve the structure. Proteins with IDRs play critical roles in complex RNA-protein interaction and condensate formation in normal development and neuronal pathology. Overall, pRBS-ID can work as a powerful, general, and amenable method to deepen our knowledge of the dynamic interactions between proteins and nucleic acids.”

In summary, the manuscript in its current form, due to the limited mechanistic and biological insights, does not warrant publication in Nature Communication but is more suited for specialized journals like RNA or RNA Biology.

- The focus (and strength) of our work is to provide a methodological platform and resources. This method is highly amenable and can be combined with other methods to address a wide range of biological questions. We hope that the reviewer will agree on the merits of our study.

Reviewer #2 (Remarks to the Author):

Comments for the Author

Summary

Bae and colleagues present a manuscript entitled "Photoactivatable ribonucleosides mark base-specific RNA-binding sites" which is an extension of earlier work from the same labs that used chemical removal of RNA adducts to proteins/peptides after UV cross-linking and final identification of RNA-binding sites (RBSs) in proteins. In the present study, the authors combine this with UV irradiation (365 nm) of photoactivatable nucleotides and establish a modified mass spectrometry analysis pipeline to look at the data from what is now called pRBS-ID.

General remarks

The manuscript is well written and understandable. The images are clear and the particularly the schematics help to understand the experimental approach.

Overall, the manuscript is interesting and a logical next step of the recently-published HF-based RNA cleavage protocol which was done using 254 nm UV cross-linking. The method thus extends the available repertoire of techniques to identify RBSs in proteins. The study focuses on the method itself and the presented data is chosen to validate the technique with already-known RBPs/RBSs instead of presenting novel and/or unexpected sites.

- We are grateful for the insightful and detailed feedback.

Major criticism

My biggest concern is the presentation of the method in comparison to other protocols and how the authors justify their "p" in pRBS-ID.

- First of all, the manuscript states that UVC (254 nm) cross-linking results in uridines (U) being cross-linked exclusively. While it is true that U is found as the nucleotide that cross-links most efficiently, the cited studies (as well as other literature) do not support exclusive cross-linking with U.

- as a result of this, the reasoning of why we need a new method that goes beyond "U cross-linking" is not really a good reason d'être for pRBS-ID

- We appreciate the reviewer's comment and apologize for not properly presenting the earlier contributions. As pointed out by the reviewer, UVC not only crosslinks proteins to U but also to other bases such as G (albeit at very low frequencies). The G-crosslinks have been captured in previous CLIP-seq-like studies, where the crosslinked RNA molecules can be amplified via PCR for detection by deep sequencing (well-analyzed in Feng et al., *Mol. Cell*, 2019 paper from Chaolin Zhang's group).

However, to our knowledge, the non-U crosslinks haven't yet been captured by mass spectrometric approaches. This applies to both nuclease-based (RNPxl, iTRAPP, CAPRI, CLIR-MS/MS) and HF-based (RBS-ID) approaches. This was likely because the amount of non-U-crosslinks generated by UVC irradiation was too small to bring the corresponding peptides over the detection limit in LC-MS. Unlike sequencing-based methods, there are currently no technologies available to amplify the small amounts of crosslinked peptides for detection. Hence, the identification and characterization of non-U modification masses with UVC-crosslinks has remained highly challenging.

pRBS-ID was able to overcome this hurdle by utilizing photoactivatable ribonucleosides (PARs; 4SU, 6SG) which have been shown to crosslink to proteins more efficiently than their respective natural RNA base counterparts (U and G; Hafner et al., *Cell*, 2010). In particular, metabolic labeling of 6SG followed by UVA-crosslinking allowed us to bring the G-crosslinked peptides over the detection limit in LC-MS. Although the numbers

were small, to our knowledge, our work is the first report of non-U crosslinks identified by mass spectrometry.

In addition to expanding the base repertoire, pRBS-ID offers many advantages over the previous methods. PARs can be used for metabolic pulse-labeling to capture distinct temporal stages of RNP remodelling. pRBS-ID also can provide confident PAR-binding evidence that can facilitate functional studies. Moreover, when using purified proteins *in vitro*, one can obtain a dense interaction map of RBS on a protein of interest, which can be combined with structural and biochemical data.

We regret that our original description was rather vague and unclear. Thus we revised the manuscript to clarify these points in lines 56-76 and lines 313-327 of the Introduction and Discussion sections, respectively.

Lines 56-76:

“The original protocol of RBS-ID was developed to map UVC-crosslinked RBSs. UVC (254 nm) has been widely utilized to capture RNA-protein interactions, yet it holds substantial methodological drawbacks. UVC irradiation induces a covalent bond mostly between Us and amino acids in close proximity while the other RNA bases (A, G, or C) are crosslinked at low frequencies¹⁶. The crosslinked non-U bases could be PCR-amplified and detected by sequencing¹⁶, but the peptides cannot be amplified so they remain below the detection limit of LC-MS/MS⁹⁻¹⁵. Thus, RBS identification using UVC-crosslinking has been limited to U-interactions, leaving out the other bases in RNA. Moreover, the low efficiency of UVC-crosslinking¹⁷ made it difficult to identify low-abundance RNA-protein interactions from small amounts of samples. Thus, although RBS-ID could drastically expand the RBS maps, its application is still limited to abundant proteins, U-base, and amino acid types sensitive to UVC irradiation.

Photoactivatable ribonucleoside (PAR)-specific UVA-crosslinking is a promising alternative that can relieve the aforementioned shortcomings¹⁷⁻²¹. UVA irradiation (365 nm) selectively activates RNA-incorporated PARs to crosslink RBPs. The most widely utilized PARs are 4-thiouridine (4SU) and 6-thioguanosine (6SG)¹⁷⁻²¹, which can expand the base-interaction repertoire of RBSs from single to double base types (U and G) with higher crosslinking efficiencies compared to their respective natural RNA base counterparts¹⁷. Selective PAR labeling and efficient UVA-crosslinking would be particularly useful for the

mass spectrometric detection and identification of peptides crosslinked to individual RNA base types even at low abundances.”

Lines 313-327:

“pRBS-ID can be used as a widely applicable and amenable platform for investigating nucleic acid-protein interactions. When combined with pulse labeling and specific RNA probes, the method can profile RBSs in proteins bound to specific RNA types at various temporal stages. For instance, RBSs in proteins bound to viral genomic RNA or viral subgenomic RNAs can be selectively captured through metabolic labeling at a particular stage of the viral life cycle. In addition, RBPs acting at different stages of RNA lifetime can be distinctively captured via pRBS-ID. Importantly, the PAR-adducts provide direct evidence for RNA binding, thereby offering strong confidence in the identification. Furthermore, the single amino acid-resolution of RBS mapping can be of great value in unraveling the action mechanisms of RBPs.”

- to make things worse, the authors then use 4SU-based cross-linking at 365 nm which truly is an exclusive U cross-link (only 4SU-modified nucleotides can form cross-links) but they call the 254 nm cross-link “U-RBS” (Fig. 3). This is not only confusing but misleading in my opinion and I strongly recommend to reconsider the labeling of the UVA samples.

- We agree with the reviewer that the usage of the term “U-RBS” (which was to refer to UVC-crosslinks) was confusing. To resolve this issue, we changed “U-RBS” into “UVC-RBS” throughout the manuscript. This way, the RBSs crosslinked via UVC would be clearly distinguished from UVA (4SU-RBS).

- Another issue is how the authors describe work of others to claim that their new approach is necessary. While I agree that HF-based removal of RNA is superior to enzymatic cleavage for the described reasons, the authors make unfair comparisons with older methods:

- l. 72ff: The authors suggest that RNPxl and TRAPP have only had "limited success" in identifying RBSs. A rather bold statement for a study that finds 70 sites using the 6SG nucleotide (which is in the ballpark of the criticised papers).

- Thank you for raising this concern. We regret that our initial manuscript was impertinent in describing previous contributions. In light of the reviewer's comments, we revised the text to acknowledge the advances previous studies have brought into this field as in lines 76-81.

Lines 76-81:

"Two previous studies attempted to map RBSs crosslinked to 4SU using nucleases to cleave the RNA moiety crosslinked to peptides in yeast cells^{9,12}, bringing many interesting insights to the molecular interactions between RNA and proteins."

- l. 76 ff: The authors claim that their approach is much more efficient in terms of sample size while comparing HeLa cell culture to yeast. This is like comparing apples and pies as yeast as a liquid (and turbid) culture is way harder to cross-link than a monolayer cell line in a dish. Thus, the large input amount for yeast is most likely a result of poor UV cross-linking than of the downstream approach. I recommend to find a better comparison, or to support your claims by providing experimental data of pRBS-ID from yeast.

- We appreciate the criticism and agree with the reviewer that the comparison of datasets from yeast and HeLa cells is unfair. Hence, we removed the comparison (Fig. 2e in the original manuscript) from our revised manuscript. Because there is no nuclease-based 4SU-RBS dataset from human cells available for comparison, and we do not have experience with yeast cell culture, it is not possible for us to provide experimental data from yeast within a reasonable time. Please also note that the benefits of chemical RNA digestion in RBS identification had already been shown previously (Bae et al., NSMB, 2020).

- The large difference between U and G sites (after 4SU or 6SG incorporation, respectively) is worrisome, particularly since the authors describe the need to have a method that does not only focus on U (see comment above). I agree however that this is a useful feature and, as proof-of-principle, showing the potential of pRBS-ID. However, one concern is how much statistical data can really be derived from such a rather small dataset

- We appreciate the reviewer's point that the limitation of this work is the low coverage of 6SG-RBSs, largely owing to the inefficient crosslink nature of 6SG bases (as also witnessed in previous PAR-CLIP studies). Still, we feel that the 4SU vs. 6SG comparison holds value. We acknowledge that cautions should be made when interpreting this comparative data. Thus, we discussed the caveat of the low 6SG coverage and the potential rooms for improvement in the Discussion section (lines 346-355). And we brought this limitation more upfront in the Results section regarding Fig. 3 of the revised manuscript (lines 245-248).

Lines 346-355:

"Limitations of the current study include the low coverage of 6SG-RBSs owing to its inefficient crosslinking efficiency. This can be improved by increasing the input amount or using enhanced separation technologies prior to the MS/MS analysis. Thus, for general purposes to identify and profile RBPs and RBSs, 4SU would be more useful than 6SG. However, for RBPs that interact specifically with G residues, 6SG would be a highly valuable tool, especially when applying this method to purified or enriched proteins of interest."

Lines 245-248:

"Next, we analyzed the differences in RBSs crosslinked to distinct base types (U or G) using the 4SU-RBS and 6SG-RBS datasets. Although the number of identified 6SG-RBSs may be rather small to make highly accurate comparisons, the analysis can serve as a proof-of-principle to dissect the base-specific features of RNA-protein interactions."

- Fig. 3 c/d only compares 254 with 4SU. It would still be interesting if the "G cross-links" are a subset of the UVA-found RBSs or rather novel.

- The comparison of 4SU-RBS and 6SG-RBS can be found at Fig. 2c. Although the number is small, the 6SG-RBS does add onto the 4SU-RBS dataset.

Fig. 2c

Fig. 2c. Comparison of identified 4SU-RBSs and 6SG-RBSs (left) or the respective protein groups (right).

- For Fig. 3f, my question is how statistically sound the analysis can be for 70 sites and 20 (potential) amino acids. Can you provide the frequencies of the other 15 (if only for this review and not as part of the paper) of 3f and 3g?

- As suggested, we provide the frequencies of all 20 amino acids in Supplementary Fig. 8. The exact values are also available in the Source Data file.

Supplementary Fig. 8

Supplementary Fig. 8. Frequencies of 20 amino acids in RBS datasets.

a-c Amino acid frequencies of UVC-RBS (a), 4SU-RBS (b), or 6SG-RBS (c), respectively normalized by those of all sequences in proteins where each RBS type was identified.

- Finally, I am missing a discussion of the limitations of the method. Hydrogen fluoride is not at all easy to handle (toxicity) and not each lab in the field has been able to take up the approach due to this. I am sure that the authors can point out other potential issues that help the reader to make an informed call when pRBS-ID is the best method to pick up and when other protocols are superior.

- This is a very good suggestion. We added a separate paragraph for limitations of the study at the end of the Discussion section to describe the toxicity of HF, along with the potential need to find safer alternative chemicals that can completely digest peptide-crosslinked RNA (lines 352-355). Also, we emphasize this safety issue in the Methods section (lines 402-405).

Lines 352-355:

"In addition, we would like to call attention to the strong toxicity of HF which should be handled with extreme caution and inside a fume hood³⁸. Finding safer alternative chemicals with similar hydrolyzing activity would greatly help resolve the issue and broaden the usage of chemical RNA digestion to map RBSs."

Lines 402-405:

"All experimental steps exposing HF to air were carried out inside a fume hood with extreme caution. Calgonate® gels (Calgonate) were kept readily available for emergency use, and all procedures followed the safety measures provided by the manufacturer."

Minor points

- Fig 4: The plots lack y-axis markings. Despite the values being artificial units, it would be great to have some graphical cues to be able to better compare the intensities of different green/grey bars in one protein.

- We added y-axis markings to Fig. 4 as shown below. Please note that the exact intensity values are present in the Source Data file.

Fig. 4

Fig. 4. Base-specific RNA-protein interactions in RNP structures

a-d RBSs identified in this study (top) and their positions in known structures (bottom). One or two top-ranking RBSs from each protein are highlighted in green. The U and G residues proximal to RBSs are shown in blue and red, respectively. **a** Interaction between H411 or F487 in PTBP1 (4SU-RBSs) with proximal Us in structural data (PDB: 2ADC³⁰). **b** Interaction between Y298 in HNRNPF (both 4SU- and 6SG-RBS) with proximal G in structural data (PDB: 2KFY³¹). **c** Potential interaction between F49 in FAU (4SU-RBS) and most proximal U in structural data (PDB: 4V6X³³). **d** Potential interaction between W44 in EDF1 (6SG-RBS) and proximal Gs in structural data (PDB: 6ZVH³⁴).

- Results section from line 99 on: Since Nature Communications is not a mass spec-centric journal, I recommend to spend one or two sentences to quickly explain what the difference between M1 and M2 spectra is. This will render the manuscript much more approachable to a large audience to be able to follow the reasoning of the newly-established analysis pipeline.

- We thank the reviewer for the suggestion. We explain the differences between MS1 and MS2 spectra in the first paragraph of the Results section (lines 102-104).

Lines 102-104:

"The MS/MS data consists of two layers. MS1 measures the mass of a whole peptide precursor ion which is subsequently isolated in the gas phase and broken into smaller fragment ions, whose masses are measured in the MS2 stage."

- Methods: The link to the Github page which should contain the code/scripts for data analysis does not work (yet?).

- The Github link was set to private at the stage of initial submission. Now, the repository has been made public.

Reviewers' Comments:

Reviewer #2:

Remarks to the Author:

The authors addressed my concerns to my satisfaction in the revised manuscript.